# Effects of the Seed Oil of *Carica papaya* Linn on Food Consumption, Adiposity, Metabolic and Inflammatory Profile of Mice Using Hyperlipidic Diet

**DOI:** 10.3390/molecules27196705

**Published:** 2022-10-08

**Authors:** Lidiani Figueiredo Santana, Bruna Larissa Spontoni do Espirito Santo, Mariana Bento Tatara, Fábio Juliano Negrão, Júlio Croda, Flávio Macedo Alves, Wander Fernando de Oliveira Filiú, Leandro Fontoura Cavalheiro, Carlos Eduardo Domingues Nazário, Marcel Arakaki Asato, Bernardo Bacelar de Faria, Valter Aragão do Nascimento, Rita de Cássia Avellaneda Guimarães, Karine de Cássia Freitas, Priscila Aiko Hiane

**Affiliations:** 1Graduate Program in Health and Development in the Central-West Region of Brazil, Federal University of Mato Grosso do Sul (UFMS), Campo Grande 79070-900, Brazil; 2Health Science Research Laboratory, Federal University of Grande Dourados, Dourados 79804-970, Brazil; 3Oswaldo Cruz Foundation, Campo Grande 79074-460, Brazil; 4Laboratory of Botany, Institute of Biosciences, Federal University of Mato Grosso do Sul, Campo Grande 79070-900, Brazil; 5Faculty of Pharmaceutical Sciences, Food and Nutrition, Federal University of Mato Grosso do Sul (UFMS), Campo Grande 79079-900, Brazil; 6Chemistry Institute, Federal University of Mato Grosso do Sul, Campo Grande 79070-900, Brazil; 7Medical School, Federal University of Mato Grosso do Sul, Campo Grande 79070-900, Brazil; 8Diagnostic Medicine Laboratory-Scapulatempo, Campo Grande 79002-17, Brazil

**Keywords:** Brazilian fruit, oil, dyslipidemia

## Abstract

Background: Studies indicate that different parts of *Carica papaya* Linn have nutritional properties that mean it can be used as an adjuvant for the treatment of various pathologies. Methods: The fatty acid composition of the oil extracted from the seeds of *Carica papaya* Linn was evaluated by gas chromatography, and an acute toxicity test was performed. For the experiment, Swiss mice were fed a balanced or high-fat diet and supplemented with saline, soybean oil, olive oil, or papaya seed oil. Oral glucose tolerance and insulin sensitivity tests were performed. After euthanasia, adiposity, glycemia, total cholesterol and fractions, insulin, resistin, leptin, MCP-1, TNF-α, and IL-6 and the histology of the liver, pancreas, and adipose tissue were evaluated. Results: Papaya seed oil showed predominance of monounsaturated fatty acids in its composition. No changes were observed in the acute toxicity test. Had lower food intake in grams, and caloric intake and in the area of adipocytes without minimizing weight gain or adiposity and impacting the liver or pancreas. Reductions in total and non-HDL-c, LDL-c, and VLDL-c were also observed. The treatment had a hypoglycemic and protective effect on insulin resistance. Supplementation also resulted in higher leptin and lower insulin and cytokine resistance. Conclusions: Under these experimental conditions, papaya seed oil led to higher amounts of monounsaturated fatty acids and had hypocholesterolemic, hypotriglyceridemic, and hypoglycemic effects.

## 1. Introduction

America is home to 124,933 species of plants, distributed in 6227 genera and 355 families [1,2]. Brazil has approximately 1.8 million biological species; among these, about 350 thousand are cataloged. Thus, it has become a great supplier of biological material varieties that are useful for technological exploration [3] and resources that favor the identification of new therapeutic means for the treatment of various diseases [4].

In this sense, the investigation of fruits and their therapeutic applicability has gained ground due to their wide consumption and their presentation of a useful nutritional composition that includes fibers, vitamins, minerals, and phytochemical compounds (phenols, bioactive compounds, carotenoids, among others) that manifest several health effects [5].

Papaya (*Carica papaya* Linn), which was discovered in southern Mexico and northern Nicaragua, was brought to Brazil in 1587, showing excellent adaptation to the local climate and becoming the most produced and commercialized fruit, transforming Brazil into one of its main world producers, together with the mentioned countries [6].

This fruit is consumed in its natural form, as well as processed, being found in the form of jelly, sweets, and pulps. In processing, the husks and seeds are removed, resulting in a loss of approximately 50% of the product, and with that, a significant percentage of waste is generated. The larger the fruit, the greater the quantities of seeds it contains, and, according to Allan (1969) [7], a single papaya can produce about 1000 seeds or more, representing approximately 15% to 20% of the seeds (in wet weight); once the seeds are not consumed, a large amount of biomass is discarded (approximately 2.38 thousand tons per year), generating a huge amount of waste and by-products that cause environmental organic pollution [8].

Studies have shown that papaya seed extract has a bioactive component, benzyl isothiocyanate, and contains significant amounts of glucosinolates, which have an inhibiting action on the development of cancer [9] and verminosis [10] and can be used in the treatment of gastric ulcers [11], diuretic actions [12], antibacterial, hypoglycaemic, and anti-inflammatory activity [13]. α and δ-tocopherol are the predominant tocopherols, at 51.85 and 18.9 mg·kg^−1^, respectively [14]. The content of total phenolic compounds was found to be 957.60 mg·kg^−1^, and when carotenoids were quantified, β-Cryptoxanthin (4.29 mg·kg^−1^) and β-carotene (2.76 mg·kg^−1^) were found [15,16], in addition to producing a specific proteolytic enzyme called papain that has numerous health benefits [17].

In the fatty acid profile, oleic fatty acid is predominant in the seeds (71.30%), followed by palmitic (16.16%), linoleic (6.06%), and stearic fatty acids [14,15,16].

Obesity and the accompanying metabolic disorders characterize metabolic syndrome. Among the various treatments proposed, the use of products obtained from plants or fruits has been reported since antiquity, as they present high concentrations of vitamins, bioactive compounds, and a lipid composition that reduces inflammatory markers and platelet aggregation and protects against thrombogenesis and oxidative stress, in addition to preventing hypercholesterolemia and hypertriglyceridemia [18]. So, plant species represent not only a food source of nutrients, but also of substances of great interest to human health [19].

Considering the composition of papaya seeds, their use can be beneficial for the prevention and/or treatment of obesity and associated metabolic disorders; however, as yet, there is no evidence of the possible effect of papaya oil on obesity and metabolic disorders. As such, the objective of this study is to evaluate the therapeutic effects of oil from the seeds of *Carica papaya* Linn in animals receiving a high-fat diet. The study may contribute to the discovery of a new biological resource for the prevention and/or treatment of obesity and its comorbidities, as well as contributing to an increase in papaya productivity, benefiting the economy, as well as encouraging the use of discarded fruit waste.

## 2. Results

### 2.1. Fatty Acid Composition of Papaya Seed Oil (Carica papaya Linn)

In the analysis of fatty acid composition, higher concentrations of monounsaturated fatty acids (74.75%) were observed, with oleic acid predominating (C18:1ω9) at 72.93% (±0.01). Among the saturated fatty acids (20.57%), there was a predominance of palmitic acid (C16:0) at 16.95% (±0.09). These were followed by polyunsaturated fatty acids (4.16%), with higher concentrations of linoleic acid (C18:2ω6)—3.64% (±0.02) (Table 1).

### 2.2. Acute Toxicity Test and “Hippocratic Screening”

In the acute toxicity test, the seed oil of papaya (*Carica papaya* Linn), with a dose of 5000 mg/kg administered to healthy animals (Group OM), did not cause any change in the behavioral parameters analyzed compared to animals supplemented with only saline (Group CT), and there was no significant difference in food and water consumption between groups.

There were no deaths over the 14-day period (acute toxicity) of observation. The body weights of the animals, as well as the weights of the liver, spleen, kidneys, lungs, and heart, did not show significant differences when comparing the groups, and no macroscopic alterations were observed (data not shown).

### 2.3. Trial Period

#### 2.3.1. Food Intake and Caloric Consumption

In the first month of the study, the food consumption in grams of the HPL OM group was lower, showing a difference when compared to the CT group (*p* < 0.01). In the second month, the HPL OM still presented lower consumption (grams), but with a difference in relation to the CT, AIN-93, and HPL groups (*p* < 0.01). There was a difference in total consumption when compared to the CT and AIN-93 groups (*p* < 0.001); that is, throughout the experiment, the HPL OM group presented lower food consumption (grams) when compared to the CT and AIN-93 groups (Table 2).

Regarding caloric intake, it was observed that in the third, fifth, and eighth weeks of supplementation, the animals that received papaya seed oil had lower caloric intake, with a significant difference when compared to the CT and HPL groups (*p* < 0.001) (data not shown).

#### 2.3.2. Assessment of Body Weight, Weight Gain, Body Fat Percentage, and Adipocyte Area

At the beginning of the experiment, before commencing weight gain induction via diet and the simultaneous supplementation of the experimental groups, all animals were weighed and evenly distributed across the groups (*p* = 0.938) (Figure 1).

One week after the change in diet and supplementation, the HPL OM group (35.3 g) showed lower weight gain and a statistical difference compared to the HPL (41.8 g) and HPL OS (40.9 g) groups (*p* < 0.05). The difference between the HPL OM (36.7 g) and HPL group (43.2 g) persisted in the second week of evaluation (*p* < 0.05).

In the following weeks, in addition to total weight gain, supplementation with oil from *Carica papaya* Linn seeds did not show significant effects it did not minimize body weight gain during the experimental period.

After eight weeks of treatment, the animals were euthanized, and the sites of adipose tissue were removed. In the evaluation of weight (g) at the sites of epididymal, perirenal, retroperitoneal, mesenteric, and omental adipose tissue, the *Carica papaya* Linn seed oil showed no effects on adiposity or on body fat percentage when compared to the negative control groups (Table 3).

However, when evaluating the adipocytes area, the group that received papaya seed oil presented lower values with a statistical difference from the HPL, HPL OS, and HPL AZ groups (Figure 2 and Figure 3).

#### 2.3.3. Serum Metabolic Changes

##### Triglycerides and Cholesterol (Total and Fractions) in Serum

Total cholesterol values were lower in the HPL OM, with a statistical difference when compared to the HPL group (*p* = 0.002) (Figure 4).

When verifying the HDL cholesterol values, only the HPL OM group presented high values with a statistical difference from the AIN 93 group (*p* = 0.008). As regards non-HDL cholesterol, the HPL OM group obtained lower values, with statistical difference when compared to the HPL group (*p* < 0.001).

The triglyceride values showed no significant difference between the groups; on the other hand, lower LDL cholesterol values were found in HPL OM, with a statistical difference when compared to HPL, AIN-93, and HPL OS (*p* < 0.001). In the assessment of VLDL cholesterol, the HPL OM group presented lower values, with a statistical difference when compared to the HPL group (*p* = 0.038).

Thus, it can be seen that supplementation with *Carica papaya* Linn seed oil manifested a reduction in total cholesterol concentrations in non-HDL cholesterol, LDL cholesterol, and VLDL cholesterol, and an increase in HDL-c.

##### Glycemic Profile: Fasting Blood Glucose, Oral Glucose Tolerance Test, and Insulin Sensitivity

The animals supplemented with oil from the seeds of *Carica papaya* Linn had lower fasting blood glucose values, with statistical difference compared to HPL (*p* = 0.003) and similar values to CT and AIN-93 (Figure 5A).

When the insulin sensitivity test was carried out, the group supplemented with papaya seed oil showed lower values at all times compared to the groups receiving a high-fat diet; however, at 30 min, there was a statistical difference when compared with HPL (*p* < 0.05) (data not shown). The results of the oral glucose tolerance test and the insulin sensitivity test were confirmed by performing an analysis of the area under the curve of both tests (Figure 5B).

In the area under the curve of the oral glucose tolerance test, the HPL OM group presented a statistical difference when compared with the CT and AIN-93 groups (Figure 5C).

Therefore, papaya seed oil had a hypoglycemic effect and a protective effect on insulin resistance.

##### Adipokines Concentration: TNF-α, IL-6, MCP-1, Insulin, Resistin, and Leptin

In the evaluation of leptin concentrations, the HPL OM group showed high values, with statistical difference in relation to CT, HPL, HPL OS, and HPL AZ (*p* < 0.001) (Figure 6A).

In the quantification of insulin and resistin concentrations, HPL OM obtained lower and significantly different values from the HPL AZ group (*p* < 0.001).

When evaluating the concentrations of IL-6, MCP-1, and TNF-α, the group supplemented with papaya seed oil showed lower values and a statistical difference when compared with the HPL OS group (*p* < 0.001).

Thus, supplementation with oil from *Carica papaya* Linn seeds increased leptin concentrations, while the values of resistin and insulin remained similar to those of the other groups, with the exception of HPL AZ.

#### 2.3.4. Histological Analysis of the Liver and Pancreas

After eight weeks of treatment, the animals were euthanized, and the liver and pancreas were removed and weighed, and then sent for histological analysis.

There was no change as regards weight, but the histological evaluation of the liver showed lower levels of steatosis in HPL OM compared to the CT and AIN-93 groups (*p* < 0.001), with the same result for microvesicular steatosis (*p* = 0.005). In ballooning, more cells were identified in the HPL OM group, with a statistical difference in relation to the CT and HPL AZ groups (*p* = 0.015). There was a lower presence of Mallory hyaline in the HPL OM when compared to CT and HPL AZ (*p* = 0.030). In the other parameters evaluated in the liver and pancreas, there was no statistical difference between the groups (Table 4).

Therefore, supplementation with oil from *Carica papaya* Linn seeds did not have significant impacts on the histological parameters of the liver and pancreas.

## 3. Discussion

Fruits such as papaya, widely available and consumed, produce a high percentage of waste, which arouses interest in the full use of the fruit, especially its seeds, which should be encouraged by more scientific studies, such as those that address biological activities, benefits, and medicinal applications of the seeds [20].

The results of our study on papaya seed oil corroborate the findings of the literature, with a high degree of unsaturation according to the refractive index [21] and the proper handling and preservation of the raw material being necessary, according to the peroxide index [22]. To verify the relatively high molecular weight and low molecular weight fatty acid content, the saponification index is used, which has identified adulteration with other oils or fats containing differently sized fatty acids, which fill them with unsaponifiable materials, such as paraffin and oil minerals [23].

However, omega 3 fatty acids are attributed a greater preventive effect than omega 6 fatty acids [24]. Omega 3 fatty acids are considered anti-thrombogenic and anti-atherogenic, because their main effect in the context of coronary heart disease is a reduction in the production of thromboxane A2 (TXA2), along with platelet aggregation that favor thrombosis, which occurs as PUFAs *n*-3 competes with AG (*n*-6) to serve as a precursor in the synthesis of eicosanoids, thus causing a change in their production. These actions promote the anti-thrombotic effect because of vasodilation and the reduced platelet aggregation [25].

The atherogenicity (AI) and thrombogenicity (TI) indices are important tools used to verify the nutritional quality of a product, since these indices indicate the material’s potential to stimulate platelet aggregation; that is, the lower the values of the AI and IT, the greater the amount of anti-atherogenic fatty acids present in a certain oil/fat and, consequently, the greater the potential for preventing the onset of coronary heart disease [26].

The omega 6 and omega 3 fatty acids compete for the same enzymes, which are involved in desaturation and chain lengthening reactions. Although these enzymes have greater affinity for the *n*-3 fatty acids, the conversion of linolenic acid into PUFAs is strongly influenced by the levels of linoleic acid in the diet; very high ratios tend to reduce the production of eicosapentanoic acid (EPA), resulting in conditions that facilitate the development of inflammatory and cardiovascular diseases [25].

The results obtained in this current study demonstrate that papaya seed oil has anti-atherogenic effects, as it presents a low atherogenicity index, thrombogenicity index, and ω6:ω3 ratio, which are satisfactory qualities.

Even though papaya seed oil has a nutritional quality that guarantees its protective and health-promoting effects, it is important to investigate whether it could be toxic. The present study’s acute toxicity tests proved that the oil did not interfere with food and water consumption, body, and vital organ weights (kidney, liver, heart, lung, and spleen), or behavioral parameters (“Hippocratic screening”) when offered in high doses, according to the OECD protocol (2008) [27].

Other evidence of papaya seed non-toxicity was found in a study assessing the effects of *C. papaya* on liver recovery after the use of drugs that induce hepatotoxicity in rodents, with liver recovery occurring and liver damage developing with an increase in antioxidant enzymes, such as superoxide dismutase (SOD), glutathione (GSH), and catalase in the liver, in addition to reductions in the enzymes AST and ALT [16,28].

Similar data on CCl4-induced nephrotoxicity were derived in Wistar rats supplemented with aqueous papaya seed extract, which showed reductions in biochemical parameters such as serum levels of uric acid, urea, and creatinine, in addition to recovery from lesions in histologically evaluated kidneys [29].

Given the above, along with the results found in other studies, *C. papaya* seed oil presents in its composition important nutrients and bioactive compounds, such as antioxidants, vitamins, and minerals with nutraceutical importance and potential health effects [10,14,16]. However, to our knowledge, there are as yet no studies that substantiate such properties, nor is there any evidence of the possible beneficial effect of papaya oil against the development of metabolic disorders.

Changing the caloric supply triggers possible weight changes and favors the development of metabolic changes [30]; further, the composition of the diet can influence food consumption, as diets with greater amounts of fat tend to promote satiety for longer, reducing the amount consumed [31]. When combined with supplementation with oils, this means that fats can further reduce consumption and modify the characteristics of fatty acids, micronutrients, and bioactive compounds, impacting final consumption [32].

During the experiment, we saw a reduction in feeding in the second month of supplementation; however, a reduction in caloric intake was also noted during the first and second experimental months, as well as in total caloric intake. In addition to the caloric load, the group that received oil from *C. papaya* seeds presented in its composition 2.1 g/100 g^−1^ of fibers and 2.6 g/100 g^−1^ of proteins [10,14], these being nutrients that can promote increased satiety and delayed gastric emptying [33].

However, this consumption did not prevent body weight gain, contrary to what was observed in the study by Campuzano-Bublitz et al. (2018) [34], in which papaya pulp supplementation (4 g) in diabetic animals for 28 days manifested a reduction when compared to the other groups. Similarly, Od-Ek et al. (2020) [35] supplemented rats fed a high-calorie diet with papaya juice at doses of 0.5 and 1.0 mL/kg of animal weight for 12 weeks and observed a significant reduction in body weight similar to that in the group that consumed a balanced diet.

All the aforementioned studies relate the values obtained in body weight to supplementation with *C. papaya*, which may be due to the scientific indications that the nutritional composition of seeds inhibits pancreatic lipase, delaying lipid digestion and consequently the absorption of fatty acids [36,37].

In the study by Lee et al. (2018) [38] of oxidative stress and antioxidant system activity, which evaluated lipid peroxidase through a biomarker called malondialdehyde (MDA), it was shown that animals that received papaya juice, regardless of dose (0, 5, and 1.0 mL/kg animal weight), showed an improved imbalance of oxidative stress generation and a greater ability to detoxify or repair damage caused by decreased MDA levels and increased antioxidant levels, similar results derived in several other studies, which have demonstrated the potent antioxidant properties of papaya [35,39,40].

Furthermore, the study by Od-Ek et al. (2020) [35] also obtained a reduction in the size of adipocytes; as such, the beneficial effects of various natural products in reducing obesity are attributed to the presence of significant amounts of bioactive compounds that have antioxidant and anti-inflammatory properties [37].

One of the main effects related to reductions in body weight concomitant with the accumulation of fat, reflected in the percentage of body fat and reductions in the circumference of adipocytes, is the reduction in total cholesterol, LDL-c, and TG levels [41].

The same mechanism may have led to the reductions in total cholesterol, LDL-c, TG, and VLDL-c levels in the present study; this similarity was also found in the study of Zetina et al. (2015) [41], wherein they evaluated the effects of aqueous extracts of papaya seeds at doses of 0, 31, 62, or 125 mg/kg of body weight for 20 days in hypercholesterolemic animals.

The studies by Lunagariya et al. (2014) [36], Manna and Jain (2015) [42], and Rochlani et al. (2017) [37] found that reductions in the serum lipid profile are influenced by the ingestion of nutrients that inhibit the intestinal absorption of dietary fat through the inhibition of pancreatic lipase activity or by the fermentation of fibers producing short-chain fatty acids (propianate) that are able to inhibit the enzyme HMG-COA, which is responsible for the synthesis of cholesterol. In the more prolonged supplementation situation of the present study, we could see reduced tissue lipid accumulation, supporting the hypothesis that *C. papaya* seed oil can reduce the extent of obesity induced by a hypercaloric diet.

The increase in body weight associated with the accumulation of adipose tissue is an indicator of several metabolic changes, and the storage of adipose tissue in the visceral region results in numerous pathophysiological changes that can favor insulin resistance to different degrees, as well as manifesting increased productions of glucose and reductions in glucose uptake by peripheral tissues, such as muscle tissue [43,44].

In the present study, *C. papaya* oil had a hypoglycemic effect that was also seen in the study by Islam et al. (2019) [45], who provided it for 21 days to animals with streptozotocin-induced diabetes, along with a methanol extract of unripe papaya fruit (doses 100 and 200 mg/kg of animal weight). The groups manifested a hypoglycemic effect, with statistical difference between them (*p* < 0.001). This was also seen in the work of Ezekwe et al. (2017) [46], who induced diabetes with streptozotocin, and treated the animals with an aqueous extract of *C. papaya* root for 21 days, which manifested a reduction in fasting hyperglycemia in diabetic animals of 30.95%.

It was also observed in the present study that, when evaluating the glucose values via the glucose tolerance test (TTOG), no reductions were seen in animals that received papaya seed oil. On the other hand, in the insulin sensitivity test (TSI), the HPL OM group presented values similar to those of CT and AIN-93, both in the assessment of glycemic levels at different times of the test and in the area under the curve, along with a statistical difference in relation to the HPL group, which generated superior results.

Agada et al. (2020) [47] investigated the in vitro and in vivo inhibitory effects of hexane, methanol, and aqueous extracts from *Carica papaya* seeds on the enzymes α-amylase and α-glucosidase. These combat hyperglycemia by inhibiting these enzymes’ digestion and carbohydrate uptake after food ingestion, thus controlling the transport of glucose in the blood, and it was found that all extracts were significantly reduced in postprandial glucose levels, resulting in behavior favorable for TTOG and TSI. It was thus concluded that the inhibition of α-amylase and α-glucosidase enzymes and the prevention of oxidative stress in postprandial hyperglycemia are possible mechanisms by which antidiabetic properties are exerted.

These changes explain why, in the histological evaluation of the liver, the presence of steatosis and microvesicular steatosis was observed in all groups that received the HPL diet regardless of treatment, as well as an alteration in the cells’ water regulation, which was impaired and caused ballooning or hydropic degeneration, with a greater number of these cells being identified in the HPL OM group. Such changes are common in the context of diets with high levels of carbohydrates and/or lipids, as well as constant changes in blood glucose values that alter the transport and storage of TG in the liver (with this not necessarily being the pathological condition) [45,46].

Another important aspect relates to the concentrations of hormones such as leptin, which is predominantly (though not exclusively) secreted by adipose tissue [48,49]. The genetic inactivation of the lep gene in adipose tissue leads to undetectable levels of circulating leptin [50]. Due to its ability to promote energy expenditure and reduce food intake in lean rats, it has been closely studies [51]. However, in early clinical trials, the long-term treatment of obese patients with supraphysiological doses of leptin has not confirmed the ability of recombinant leptin to act as an anti-obesity factor [52]

In a study by Zhao et al. (2020) [53], it was observed that animals fed a high-fat diet showed no significant changes in body weight, but they did show lower energy intake, and exhibited better glucose tolerance and insulin sensitivity along with a slight reduction in tissue inflammation adipose, with no liver changes. They also maintained a higher level of sensitivity to leptin. Behaviors similar to those observed in this study were seen, suggesting that papaya seed oil supplementation associated with a high-fat diet, in the context of a leptin-sensitive state, can increase leptin levels, and may actually reduce food intake and weight body, with effects on the production and action of insulin.

Resistin, which has the effect of blocking the central action of leptin (the induction of satiety), is a small secretory protein that plays a pleiotropic role in rodents and humans. Furthermore, mouse resistin causes IR and contributes to type 2 diabetes mellitus, while human resistin plays a role in inflammation [54]. Even without a statistical difference, the HPL OM group presented low resistin values, and in the TSI group we saw the prevention of IR, which may be related to its low concentrations.

The study by Od-Ek et al. (2020) [35] did not report any statistical difference in the concentrations of leptin and resistin between the group treated with papaya pulp juice, even receiving a high-calorie diet, with the others, thus corroborating the results found.

Similar to leptin, insulin is a hormone secreted in proportion to fat stores or adipose tissue, through the hypothalamus; it regulates energy expenditure and the mechanisms of hunger and satiety [37].

Insulin is an anabolic hormone manufactured by the pancreas, more specifically by the β cells of the islets of Langerhans, in response to plasma levels of nutrients, especially glucose. It is a polypeptide that acts on lipid metabolism, facilitating the entry of the glucose molecule into the cell interior, interacting with liver, muscle, and adipose tissue cells [55]. Insulin also helps in the development of the blood–brain barrier through receptors in the hypothalamus, interacting with Y neurons and generating greater satiety; it is considered a potent anorectic signal in the central nervous system, leading to the reduced expression of genes that regulate satiety in the hypothalamus [37].

However, according to Vugic (2020) [56], pro-inflammatory molecules are secreted not only by adipocytes, but also by cells present in the extracellular matrix of adipose tissue: endothelial cells, macrophages, fibroblasts, leukocytes, and pre-adipocytes. 

These favor insulin resistance, which refers to the inability of the adipose cell to respond to insulin due to the blocking of enzymes and molecules participating in the insulin signaling cascade. This was observed in our study, and even at lower values, it did not differ statistically from the other groups.

Metabolic disorders associated with chronic diseases and inflammation were demonstrated by a high concentration of pro-inflammatory markers such as TNF-α, MCP-1 and IL-6, and a decrease in anti-inflammatory factors such as adiponectin. A diet rich in fats has the same effect [57,58].

Papaya is a good source of antioxidant phytochemicals, such as vitamin C, carotenoids, and vitamin E, which reduce oxidative stress. With reports in the literature of its antioxidant and anti-inflammatory action, the HPL OM group showed a reduction in inflammatory markers (TNF-α, MCP-1, and IL-6) (Figure 8), although not significant, suggesting that with longer treatment this result could be different, as was found by Zetina-Equivel et al. (2015) [41] and Somanah et al. (2017) [59].

These results contradict the literature, as the oil obtained from the seeds of *Carica papaya* Linn has a fatty acid composition (oleic, palmitic, linoleic, and stearic). Studies have shown its antioxidant effect, with high amounts of carotenoids, phenolic compounds, β-carotene, and β-cryptoxanthin, which have anti-inflammatory and antioxidant properties [18,60].

Drehmer and collaborators (2019) [61] have reported that olive oil, which has a similar composition to the oil in this study, is one of the main components of the Mediterranean diet, which is famous for its ability to decrease body weight, the index of body mass and the waist and hip circumferences. Among the mechanisms potentially responsible for this reduction in body weight are the activation of β-oxidation, the induction of satiety, stimulation of energy expenditure by inducing thermogenesis in brown adipose tissue, the inhibition of adipocyte differentiation, the promotion of adipocyte apoptosis, and the increasing of lipolysis [62].

However, these effects of olive oil are only optimized when it is consumed in association with an adequate and balanced diet and physical activity, and when excessive levels of fat are included in the diet, even supplementation with olive oil cannot prevent the onset of obesity, which explains the results of our study in both the HPL OM and HPL AZ groups [61] (Figure 7).

## 4. Materials and Methods

### 4.1. Obtaining Oil from Papaya Seeds (Carica papaya Linn)

The papayas were obtained from the Supply Center (Ceasa) in the city of Campo Grande, Mato Grosso do Sul, Brazil, between the months of April and August 2018 (SisGen—A38192C). They were deposited in the CGMS/UFMS herbarium with exsiccata number 81021.

The oil from papaya seeds was extracted through mechanical cold pressing, according to the methodology of Manzano (2009) [63]. This was carried out at the RTK Industry and Processes industry, located in Brasília (DF, Brazil), facilitated by the partnership established with this research group, which is part of the CNPq Research Directory—Science and Technology of Food at UFMS.

### 4.2. Fatty Acid Profile and Indices of Nutritional Quality

The extracted crude oil was subjected to esterification according to the methodology described by Maia and Rodriguez (1993) [64]. Fatty acid methyl esters were analyzed by gas chromatography (GC-MS 2010, Shimadzu, Kyoto, Japan) to obtain their individual peaks. The equipment used a flame ionization detector (FID) and a capillary column (BPX-70, internal diameter of 0.25 mm, 30 m long, and 0.25 mm thick film). The injector and detector temperatures were 240 °C. The initial column temperature was maintained at 70 °C for 2 min and then increased to 10 °C/min until reaching 150 °C, followed by an increase to 240 °C at 5 °C/min for 5 min. Individual FAME peaks were identified by comparing their relative retention times with the FAME standard (fatty acid methyl ester) (Supelco C22, 99% pure). The calculation of fatty acid contents was performed by integrating the peak areas (area percentage), and the results are expressed in g fatty acid/g extracted oil.

To determine the atherogenicity index (AI) (Equation (1)) and thrombogenicity index (TI) (Equation (2)), we used the expressions proposed by Ulbricht and Southgate (1991) [65].
AI = C12:0 + 4 × C14:0 + C16:0 ∑ MUFA + ∑ ω6 + ∑ ω3(1)
TI = C14:0 + C16:0 + C18:0 0.5 × ∑ MUFA + 0.5 × ∑ ω6 + 3 × ∑ ω3(2)

To obtain the ω6:ω3 ratio, the ratio between ω6 concentrations and ω3 concentrations was employed, according to the study by Silva et al. (2018) [66].

### 4.3. Animals

The experimental protocol was approved by the Ethics Committee for Animal Use (Protocol n^o^. 980/20178), which is part of the International Guiding Principles for Biomedical Research Involving Animals (CIOMS), Genebra, 1985; the Universal Declaration of Animal Rights (UNESCO), Bruxelles, Belgium, 1978; and the guidelines of the National Health Institutes on the use and care of laboratory animals. The animals were kept under standard laboratory conditions (12:12 light/dark cycle, lights on at 07:00 h, 22 °C, 60% humidity, food, and water ad libitum) [67]. The experiments were carried out during the light phase between 09:00 and 14:00. The animals were placed in a polyethylene box measuring 30 × 20 × 13 (GC100) (four to five animals per box). Every effort was made to minimize animal suffering and reduce the number of animals used.

### 4.4. Acute Toxicity Test

Female Swiss mice (16.6 ± 0.96g of weight) were divided into two groups (*n* = 5): The control group received saline (1 mL/kg of body wt) and the treatment group received papaya seed oil (5000 mg/kg of body wt), with the same final volume for both groups, administraded as a single dose. After treatment, the animals were observed at 30, 60, 120, 240, 360 min, and then daily for 14 days. The presence of cognitive, neuromuscular, and physical alterations was observed, assigning a zero score to the animals which were normal and graded from 1 to 4 for the changes observed according to the intensity (Hippocratic screening), together with the daily assessment of body weight and food and water consumption. After the toxicity test, the experiment was started [68].

### 4.5. Experimental Design

The high-fat diet used in this study was based on the AIN-93M, supplemented with lard in the place of ingredients such as starch in order to increase the caloric content (adapted from Lenquiste et al., 2015 [69]—Table 5).

Swiss mice (males and adults (16.4 ± 1.22 g of body weight) were subjected to 7 days of adaptation to the new environment and were later divided into experimental groups as shown in Figure 8. The experiment began with a change in the type of feed and supplementation simultaneously, with ad libitum diet and supplementation through gavage, all groups receiving a dose of 1 mL/kg of animal weight per day for 8 consecutive weeks [35,38,70].

The feed intake was measured weekly, along with the weights of the animals, before the administration of the respective treatments [71].

After eight weeks of treatment and 8-hour fasting, the animals were anesthetized with Isofluorane^®^ and euthanized by exsanguination through the inferior vena cava. Blood samples were centrifuged at 3000 rpm for 5 min, and the serum was separated and stored a −18 °C in a biofreezer for further analysis.

Five sites of adipose tissue were removed (epididymal, retroperitoneal, perirenal, mesenteric, and omental); these were weighed, with subsequent determination of the animal’s fat content (percentage of adipose tissue in each site in relation to body weight) [72].

The liver, pancreas, and epididymal adipose tissue were weighed and fixed in a 10% formalin solution. After 24 h, the tissue was transferred to a 70% ethyl alcohol solution, where it remained until the preparation of the histological slides [73].

### 4.6. Metabolic Changes in Serum

Triglycerides, cholesterol (total and fractions), HDL-c, and blood glucose were measured according to the guidelines of the commercial kit Lab Test Diagnóstica^®^, Brazil.

The oral glucose tolerance test (OGTT) was performed four days prior to the euthanasia of animals after six hours of fasting. Fasting glucose was verified via flow rate (time 0) using a G-Tech^®^ glucometer (G-TECH Free, Infopia Co., Ltd., Anyang, South Korea). Then, the animals received a D-glucose solution (Sigma Aldrich, Duque de Caxias, Rio de Janeiro, Brazil), at 2 g/kg of body weight, by gavage. A blood glucose reading was performed 15, 30, 60, and 120 min after glucose application [47].

After two days, the animals were submitted to the insulin sensitivity test (TSI). The test was performed after the animals were fed, having received an intraperitoneal injection of regular Humulin^®^ insulin (0.75 U of insulin/kg of body weight), and caudal blood samples were taken at 0, 15, 30, and 60 min after injection. Blood glucose measurements were performed using the G-Tech^®^ glucometer (G-TECH Free, Infopia Co., Ltd., Anyang, Gyeonggi-do, South Korea) [47].

The blood glucose concentrations were recorded, and from the peak blood glucose values, the area under the curve (AUC) was calculated for each mouse, and the mean was calculated for each experimental group, in terms of both TTOG and TSI [47,74].

### 4.7. Concentration of Adipokines: TNF-α, IL-6, MCP-1, Insulin, Resistin and Leptin

The concentrations of adipokines TNF-α, IL-6, MCP-1, insulin, resistin, and leptin were measured using the commercial kit MADKMAG-71K^®^ from Merck. For this purpose, the serum was separated via centrifugation. It was vortexed for 30 s and placed in a centrifuge (5000 rpm for 10 min). Then, 10 µL of serum from each animal was distributed and stored in a 96-well plate, together with 10 µL of Assay buffer solution and 25 µL of solution containing six adipokines. Blank, standard, and control parameters were prepared according to the manufacturer’s instructions (Milliplex® MAP kit, Billerica, MA, USA). Afterwards, the plate was read on Luminex^®^ by the MAGPIX^®^ software, and concentration values were obtained in µg/mL.

### 4.8. Histological Slides and Adipocyte Area

After euthanasia, samples of the liver and pancreas were removed for histological study, having been fixed in a 10% formaldehyde solution until embedded in paraffin. Then, 7 µm-thick sections were produced in the microtome, with subsequent mounting on glass slides. For each slide, four cuts were extracted from the paraffin block and stained with HE [74]. The histological analysis of the liver was performed using Kleiner’s system (2005) [75]. In the histological analysis of the pancreas, changes were evaluated according to the method of [76].

For the analysis of the adipocyte area of the epididymal adipose tissue, images were initially captured using the LEICA DFC 495 digital camera system (Leica Microsystems, Wetzlar, Germany), integrated with the LEICA DM 5500B microscope (Leica Microsystems, Wetzlar, Germany) with 20x magnification. The images were analyzed using the LEICA Application Suite software, version 4.0 (Leica Microsystems, Wetzlar, Germany), and the mean area of 100 adipocytes per sample was determined [77].

### 4.9. Statistical Analysis

The results are expressed as mean ± standard deviation (SD) for parametric data and have been analyzed using Prisma 5.0 software (GraphPad Software, Inc., San Diego, CA, USA).

Student’s t-test was used for the evaluation between two groups, and for comparisons with more than two groups, analysis of variance (ANOVA) was used to derive parametric data, and Tukey’s post test to derive non-parametric data. Values of *p* ≤ 0.05 were considered statistically significant.

## 5. Conclusions

*Carica papaya* Linn seed oil showed a composition predominantly containing monounsaturated fatty acids. The acute toxicity model showed no changes, indicating the safety of its consumption.

In our experiment offering a high-fat diet, the group supplemented with papaya seed oil showed hypocholesterolemic, hypotriglyceridemic, and hypoglycemic effects but displayed no significant signs of histopathological protection in the liver and pancreas.

No effects were seen in terms of satiety (leptin) and insulin resistance (insulin and resistin), or in inflammatory markers (tumor necrosis factor-α, monocyte chemotactic protein-1, and interleukin-6).

## Figures and Tables

**Figure 1 molecules-27-06705-f001:**
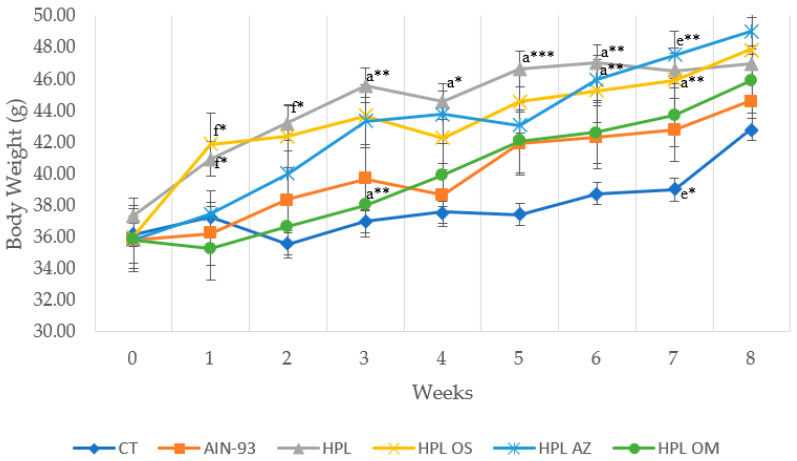
Body weight (g) of animals fed a balanced or high-fat diet and supplemented with saline, soybean oil, olive oil, or seed oil of papaya (*Carica papaya* Linn). Values are expressed as mean ± standard deviation. The letters indicate statistical difference as follows: ^a^ compared to CT; ^e^ compared to HPL AZ; ^f^ compared to HPL OM. * *p* < 0.05; ** *p* < 0.01; *** *p* = 0.004. *n* = 14–18. ANOVA followed by Tukey post test.

**Figure 2 molecules-27-06705-f002:**
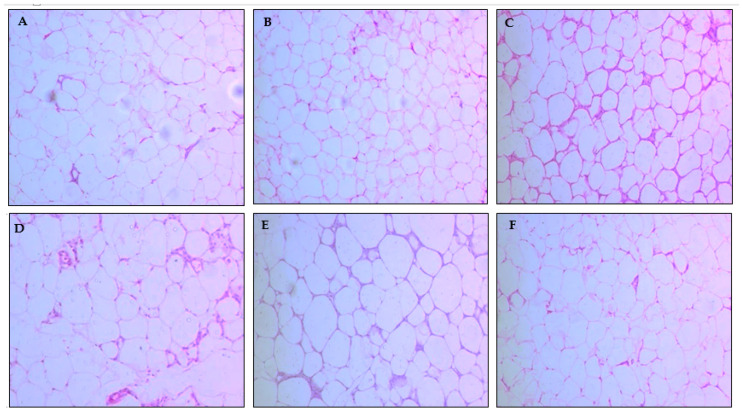
Histopathological analysis of the adipose tissue via hematoxylin and eosin (H&E—200X) for animals fed a balanced or high-fat diet and supplemented with saline, soybean oil, olive oil, or seed oil of papaya (*Carica papaya* Linn). Bar scale: 100 μm. (**A**) CT; (**B**) AIN-93; (**C**) HPL; (**D**) HPL OS; (**E**) HPL AZ; and (**F**) HPL OM.

**Figure 3 molecules-27-06705-f003:**
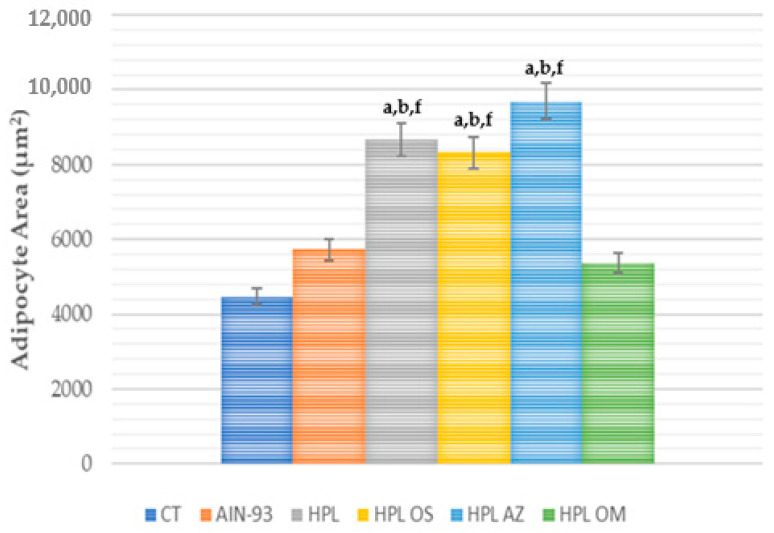
Adipocyte area (μm^2^) of animals fed a balanced or high-fat diet and supplemented with saline, soybean oil, olive oil or seed oil of papaya (*Carica papaya* Linn). Values expressed as mean ± standard deviation. The letters indicate statistical difference as follows: ^a^ compared to CT; ^b^ compared to AIN 93; ^f^ compared to HPL OM. *p* < 0.05. *n* = 6–11. ANOVA followed by Tukey post test.

**Figure 4 molecules-27-06705-f004:**
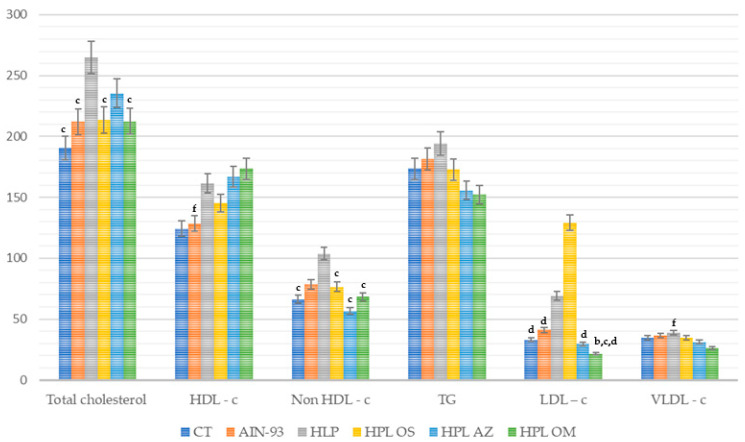
Total cholesterol, fractions, and triglycerides (mg/dL) of animals fed a balanced or high-fat diet and supplemented with saline, soybean oil, olive oil, or seed oil of papaya (*Carica papaya* Linn). Values expressed as mean ± standard deviation. Letters indicate statistical difference as follows: ^b^ compared to AIN 93; ^c^ compared to HPL; ^d^ compared to HPL OS; ^f^ compared to HPL OM. *p* < 0.01. *n* = 14–18. ANOVA followed by Tukey post test.

**Figure 5 molecules-27-06705-f005:**
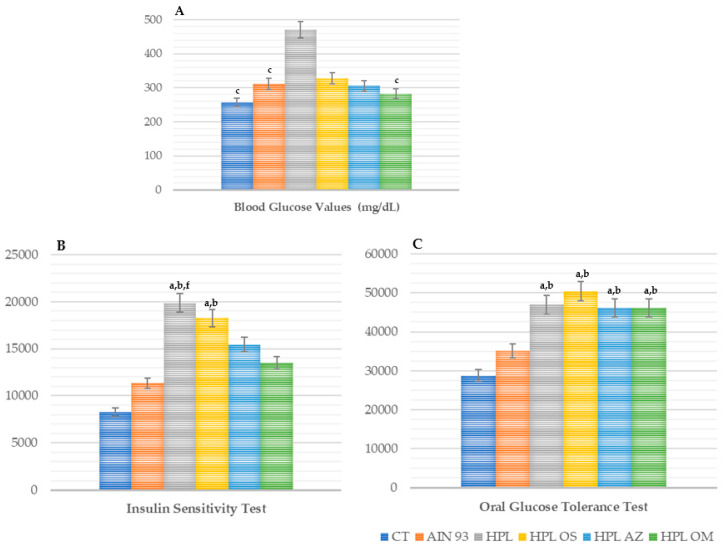
Blood glucose values (mg/dL) (**A**), area under the curve of the insulin sensitivity test (%) (**B**), and oral glucose tolerance test (%) (**C**) of animals fed a balanced or high-fat diet and supplemented with saline, soybean oil, olive oil, or seed oil of papaya (*Carica papaya* Linn). Values expressed as mean ± standard deviation. The letters indicate statistical difference as follows: ^a^ compared to CT; ^b^ compared to AIN 93; ^c^ refers to the oral glucose tolerance test (%) ^f^ compared to HPL OM. *p* < 0.001. *n* = 14–18. ANOVA followed by Tukey post test.

**Figure 6 molecules-27-06705-f006:**
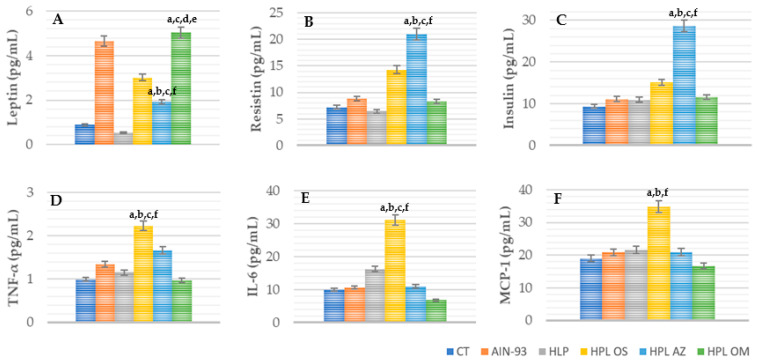
Leptin (**A**), resistin (**B**), insulin (**C**), TNF-α (**D**), IL-6 (**E**), and MCP-1 (**F**) (pg/mL) of animals fed a balanced or high-fat diet supplemented with saline, soybean oil, olive oil, or seed oil of papaya (*Carica papaya* Linn). Values expressed as mean ± standard deviation. The letters indicate statistical difference as follows: ^a^ compared to CT; ^b^ compared to AIN 93; ^c^ compared to HPL; ^d^ compared to HPL OS; ^e^ compared to HPL AZ; ^f^ compared to HPL OM. *p* < 0.001. *n* = 6–11. ANOVA followed by Tukey post test.

**Figure 7 molecules-27-06705-f007:**
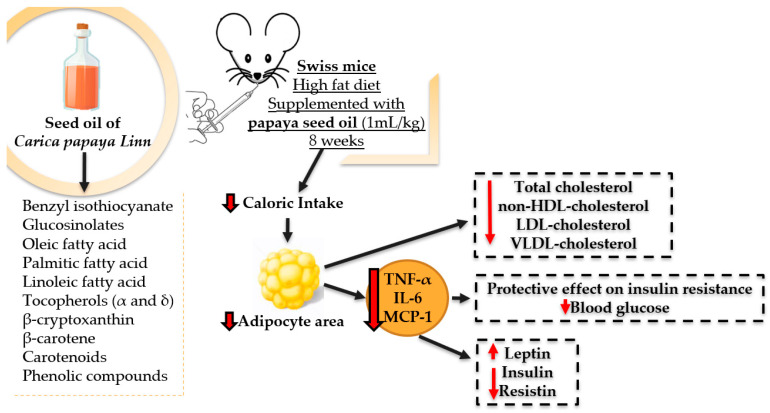
Effects of papaya seed oil supplementation in Swiss mice fed a hyperlipidic diet on food intake, adiposity, metabolic, and inflammatory profile.

**Figure 8 molecules-27-06705-f008:**
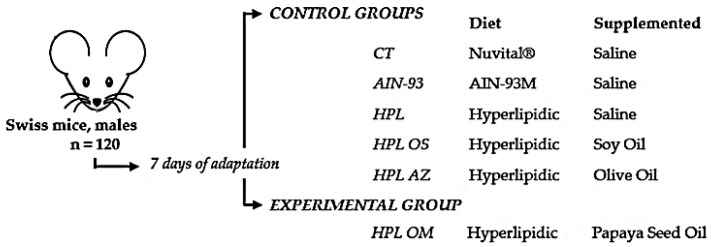
Experimental model design using a high-fat diet.

**Table 1 molecules-27-06705-t001:** Fatty acid composition of *Carica papaya* Linn seed oil.

Fatty Acids	*Carica papaya* Linn Seed Oil (%) *
Palmitic acid (C16:0)	16.95 ± 0.09
Stearic acid (C18:0)	3.62 ± 0.01
Σ SATURATED (SFA)	20.57
Palmitoleic acid (C16:1ω7)	0.34 ± 0.01
Vacenic acid (C18:1ω7)	0.87 ± 0.01
Oleic acid (C18:1ω9)	72.93 ± 0.01
Gadoleic acid (C20:1ω9)	0.33 ± 0.01
Erucic acid (C22:1ω9)	0.25 ± 0.01
Σ MONOUNSATURATED (MUFA)	74.75
Linoleic acid (C18:2ω6)	3.64 ± 0.02
Alpha-linolenic acid (C18:3ω3)	0.31 ± 0.01
Eicosadienoic acid (C20:2ω6)	0.21 ± 0.01
Σ POLY-UNSATURATED (PUFA)	4.16
Atherogenicity index	0.78
Thrombogenicity index	0.01
ω6:ω3	12.42

* Data presented as mean ± standard deviation.

**Table 2 molecules-27-06705-t002:** Food consumption (grams) and caloric intake (calories) in different periods of the experiment (first month, second month, and total period) of animals fed a balanced or high-fat diet supplemented with saline, soybean oil, olive oil, or seed oil of papaya (*Carica papaya* Linn).

Groups	Food Intake (g)			Caloric Consumption (cal)		
	First Month	Second Month	Total	First Month	Second Month	Total
CT	170.59 ± 1.80	153.71 ± 6.19	324.30 ± 24.25	743.80 ± 10.50	670.20 ± 35.50	1414.00 ± 37.30
AIN-93	128.24 ± 3.34	119.48 ± 3.98	247.72 ± 18.06 ^a,^*	487.40 ± 14.90 ^a^	454.10 ± 19.10 ^a,c^	941.50 ± 20.20 ^a^
HPL	104.99 ± 1.81	107.37 ± 2.08 ^a,^*	212.36 ± 14.61 ^a,^*	556.50 ± 12.40 ^a^	569.10 ± 15.70 ^a^	1125.60 ± 14.80 ^a^
HPL OS	92.53 ± 0.66	95.87 ± 1.76 ^a,^*	188.40 ± 12.87 ^a,b,^**	490.50 ± 5.10 ^a^	508.20 ± 14.4 ^a,c^	998.70 ± 13.70 ^a^
HPL AZ	84.17 ± 2.03 ^a,^*	85.50 ± 2.14 ^a,^*	169.67 ± 11.73 ^a,b,^**	446.20 ± 15.10 ^a,c^	453.30 ± 18.40 ^a,c^	899.50 ± 18.00 ^a,c^
HPL OM	87.07 ± 2.28 ^a,^*	81.91 ± 1.47 ^a,b,c,^*	168.98 ± 12.24 ^a,b,^**	461.50 ± 15.20 ^a,c^	434.20 ± 14.60 ^a,c^	895.70 ± 11.90 ^a,c^

Values expressed as mean ± standard deviation. Letters in the same column indicate statistical difference as follows: ^a^ compared to CT; ^b^ compared to AIN 93; ^c^ compared to HPL * *p* < 0.01; ** *p* < 0.001. *n* = 14–18. ANOVA followed by Tukey post test.

**Table 3 molecules-27-06705-t003:** Weight of adipose tissue sites and percentage of body fat of animals fed a balanced or high-fat diet and supplemented with saline, soybean oil, olive oil, or seed oil of papaya (*Carica papaya* Linn).

Groups	Adipose Tissue Sites (mg)					% Body Fat
	Epididymal	Perirenal	Mesenteric	Retroperitoneal	Omental	
CT	1,275.42 ± 814.65	158.72 ± 127.02	537.25 ± 342.19 ^e^	410.95 ± 246.33	33.92 ± 15.85 ^e^	5.63 ^d,e^
AIN-93	1584.08 ± 646.90	252.40 ± 184.67	748.53 ± 375.13	607.79 ± 270.90	31.00 ± 15.61 ^e^	7.52
HPL	1748.35 ± 742.81	322.37 ± 156.38	1,002.62 ± 613.01	612.06 ± 337.03	36.41 ± 20.84 ^e^	8.56
HPL OS	2319.62 ± 831.44 ^a^	284.56 ± 103.50 ^a^	1,114.27 ± 430.42	799.10 ± 251.65 ^a^	46.96 ± 14.97	10.65
HPL AZ	2122.54 ± 708.52 ^a^	323.46 ± 152.15 ^a^	1,124.91 ± 490.63	788.41 ± 255.67 ^a^	62.10 ± 27.74	10.32
HPL OM	2305.33 ± 722.87 ^a^	270.70 ± 96.15 ^a^	862.75 ± 334.07	719.45 ± 215.80 ^a^	41.25 ± 13.43 ^e^	7.37

Values expressed as mean ± standard deviation. Letters in the same column letters indicate statistical difference as follows: ^a^ compared to CT; ^d^ compared to OS; ^e^ compared to AZ. *p* < 0.001. *n* = 14–18. ANOVA followed by Tukey post test.

**Table 4 molecules-27-06705-t004:** Histological evaluation of the livers and pancreas of animals fed a balanced or high-fat diet and supplemented with saline, soybean oil, olive oil, or seed oil of papaya (*Carica papaya* Linn).

Evaluated Parameters		CT		AIN-93		HPL		HPL OS		HPL AZ		HPL OM		*p*
		N	%	N	%	*n*	%	*n*	%	*n*	%	*n*	%	
**Steatosis**	Up to 5%	12	100.0	9	69.2	1	8.3	2	14.3	2	14.3	1	7.1	<0.001
	5 to 33%	-	-	4	30.8	2	16.7	4	28.6	2	14.3	2	14.3	
	33 to 66%	-	-	-	-	3	25.0	4	28.6	5	35.7	7	50.0	
	More than 66%	-	-	-	-	6	50.0 ^a,b^	4	28.6 ^a,b^	5	35.7 ^a,b^	4	28.6 ^a,b^	
	Total	12	100.0	13	100.0	12	100.0	14	100.0	14	100.0	14	100.0	
**Microvesicular Steatosis**	Absent	12	100.0	11	84.6	6	50.0	7	50.0	6	42.9	7	50.0	0.005
	Presence	-	-	2	15.4	6	50.0 ^a,b^	7	50.0 ^a,b^	8	57.1 ^a,b^	7	50.0 ^a,b^	
	Total	12	100.0	13	100.0	12	100.0	14	100.0	14	100.0	14	100.0	
**Lobular Inflammation**	Absent	-	-	-	-	1	8.3	2	14.3	-	-	-	-	0.443
	<2 focuses per 200x field	12	100.0	12	92.3	9	75.0	11	78.6	13	92.9	11	78.6	
	2–4 focuses per 200x field	-	-	1	12.5	2	16.7	1	7.1	1	7.1	3	21.4	
	Total	12	100.0	13	100.0	12	100.0	14	100.0	14	100.0	14	100.0	
**Ballooning**	Absent	2	16.7	1	7.7	1	8.3	-	-	1	7.1	-	-	0.015
	Few cells	10	83.3	8	61.5	8	66.7	8	57.1	13	92.9	13	92.9	
	Many cells	-	-	4	30.8 ^a,e^	3	25.0 ^a,e^	6	42.9 ^a,e^	-	-	1	7.1 ^a,e^	
	Total	12	100.0	13	100.0	12	100.0	14	100.0	14	100.0	14	100.0	
**Mallory’s Hyaline**	Absent	12	100.0	9	69.2	10	83.3	8	57.1	13	92.9	13	92.9	0.030
	Presence	-	-	4	30.8 ^a,d^	2	16.7 ^a,d^	6	42.9	1	7.1 ^a,d^	1	7.1 ^a,d^	
	Total	12	100.0	13	100.0	12	100.0	14	100.0	14	100.0	14	100.0	
**Apoptosis**	Absent	12	100.0	13	100.0	10	83.3	13	92.9	11	78.6	13	92.9	0.305
	Presence	-	-	-	-	2	16.7	1	7.1	3	21.4	1	7.1	
	Total	12	100.0	13	100.0	12	100.0	14	100.0	14	100.0	14	100.0	
**Glycogenated core**	None/rare	12	100.0	13	100.0	12	100.0	14	100.0	14	100.0	14	100.0	--
	Some	-	-	-	-	-	-	-	-	-	-	-	-	
	Total	12	100.0	13	100.0	12	100.0	14	100.0	14	100.0	14	100.0	
**Islet of Langerhans**	Atrophy /hypotrophy	-	-	-	-	-	-	-	-	-	-	-	-	--
	Hypertrophy	-	-	-	-	-	-	-	-	-	-	-	-	
	Normal	12	100.0	13	100.0	12	100.0	14	100.0	14	100.0	14	100.0	
	Total	12	100.0	13	100.0	12	100.0	14	100.0	14	100.0	14	100.0	
**Pancreatic Acini**	Dilated	-	-	-	-	-	-	-	-	-	-	-	-	--
	Necrosis	-	-	-	-	-	-	-	-	-	-	-	-	
	Normal	12	100.0	13	100.0	12	100.0	14	100.0	14	100.0	14	100.0	
	Total	12	100.0	13	100.0	12	100.0	14	100.0	14	100.0	14	100.0	
**Inflammatory Cells**	Insulitis	-	-	-	-	-	-	-	-	-	-	-	-	--
	Perinsulitis	-	-	-	-	-	-	-	-	-	-	-	-	
	Absent	12	100.0	13	100.0	12	100.0	14	100.0	14	100.0	14	100.0	
	Total	12	100.0	13	100.0	12	100.0	14	100.0	14	100.0	14	100.0	

Values expressed in percentages. Letters in the same column indicate statistical difference as follows: ^a^ compared to CT; ^b^ compared to AIN 93; ^d^ compared to HPL OS; ^e^ compared to HPL AZ. *n* = 12–14. Pearson chi-square test, post test extension of Fisher’s exact test.

**Table 5 molecules-27-06705-t005:** Composition of experimental diets (g per kg of feed).

Ingredients (g/kg)	AIN-93M	Nuvital^®^	Hypercaloric
Starch	620.692	725.67	320.692
Casein (≥82% protein)	140.00	40.00	140.00
DL-methionine	-	100.00	--
Lard	-	-	320.00
Sugar	100.00	-	100.00
Soy oil	40.00	40.00	20.00
Cellulose	50.00	100.00	50.00
Mineral mix *	35.00	35.00	35.00
Vitamins mix **	10.00	10.00	10.00
L-cystine	1.80	1.80	1.80
Choline bitartrate	2.50	2.50	2.50
Tertbutyl hydroquinone	0.008	0.008	0.008
**Energy (cal/kg)**	**3802.80**	**3422.68**	**5302.80**
Carbohydrates (%)	75.81%	75.75%	31.73%
Proteins (%)	14.73%	16.00%	10.56%
Lipids (%)	9.47%	8.25%	57.71%
**Calories/g diet**	**3.80**	**3.42**	**5.30**

* Vitamins and ** Minerals present in the mix are in accordance with AIN-93M.

## Data Availability

Not applicable.

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
