# Peer review of "Effects of the Seed Oil of Carica papaya Linn on Food Consumption, Adiposity, Metabolic and Inflammatory Profile of Mice Using Hyperlipidic Diet"

_molecules, 2022, doi:10.3390/molecules27196705_

Round 1

Reviewer 1 Report (Previous Reviewer 1)

Santana et al. offer a revised version of the manuscript titled “Effects of the Seed Oil of Carica papaya Linn on Food Consumption, Adiposity, Metabolic and Inflammatory Profile of Mice using Hyperlipidic Diet”. Specific comments are as follows:

1) “The most important and missing item of information is the caloric content of each diet that was provided to the animals. It is very possible that a lower caloric content is present in the papaya seed oil diet, which could then translate into lower caloric intake. Are the oil-supplemented high-fat diets isocaloric? If not, the study is questionable on the effects the oil exhibits when compared with other oils.

Line 622 (Table 5). We have inserted a table describing the composition of the diets used in the study. It is possible to observe the energy value of each one per gram of diet, confirming the higher energy value in high-fat diets.”

>>>The original comment was not addressed. Are the high-fat diets (with soy, olive, and papaya oils) isocaloric?

2) “Also, parts of Figures 2 and 3, and Table 4 are off the page. These issues may be from the formatting by the journal, but this should have been caught by the authors before approval.

Done. We made the corrections.”

>>>Figure 2 and Table 4 are still off the page. It is recommended that the authors very carefully examine the pdf of the manuscript before approval within the submission system.

3) As noted before, stating that animals are +/- 16g makes no sense. Provide the average mass of the mice and the appropriate standard deviation.

4) Again, how do the atherogenicity and thrombogenicity indices compare to the other diets in the current study? Provide a discussion within the manuscript.

5) This reviewer may have missed this in the first submission, but there are no units provided for Figure 5B and 5C.

6) Some comments by the other reviewer could benefit with discussion points in the manuscript.

Author Response

Dear Reviewer 1,

We thank you for spending your time working in our manuscript. We know that these reviews are time demanding and your suggestions improved our article review.

We have modified our article according to your considerations.

The changes are listed below and marked using the “Track Changes” highlighted in green in the file.

1) “The most important and missing item of information is the caloric content of each diet that was provided to the animals. It is very possible that a lower caloric content is present in the papaya seed oil diet, which could then translate into lower caloric intake. Are the oil-supplemented high-fat diets isocaloric? If not, the study is questionable on the effects the oil exhibits when compared with other oils. 

Line 622 (Table 5). We have inserted a table describing the composition of the diets used in the study. It is possible to observe the energy value of each one per gram of diet, confirming the higher energy value in high-fat diets.”

 >>>The original comment was not addressed. Are the high-fat diets (with soy, olive, and papaya oils) isocaloric?

Dear reviewer,

We are sorry for not having satisfactorily clarified the previous question. It is important to emphasize that all animals that were part of the groups that received supplementation via gavage once a day of soybean oil, olive oil or papaya seed oil, were fed the same type of diet ad libitum, that is, the hypercaloric diet described in table 5. The hypercaloric diet used contains 5.30 calories per gram of diet, while the nuvital control diet contained 3.42 calories per gram of diet and the AIN-93M contained 3.8 calories per gram of diet. Also, as a way of ensuring that there would be no difference in energy supply between the groups, the volume of oil administered via gavage to the three groups and study was the same (1 mL per kilogram of animal weight).

2) “Also, parts of Figures 2 and 3, and Table 4 are off the page. These issues may be from the formatting by the journal, but this should have been caught by the authors before approval.

Done. We made the corrections.”

>>>Figure 2 and Table 4 are still off the page. It is recommended that the authors very carefully examine the pdf of the manuscript before approval within the submission system.

Dear reviewer,

We are really concerned about this text distortion. It is important to mention that we prepared the manuscript according to the instructions observed on the Instructions for Authors page (https://www.mdpi.com/journal/molecules/instructions), and at the time of submitting the text, it is not unformatted. However, after submission to the system, the text becomes misaligned. As a way of looking for a definitive solution, we are writing to the editor to report this issue and we hope he can help us. Once again, we are sorry for causing such inconvenience, but we are looking for a permanent solution.

3) As noted before, stating that animals are +/- 16g makes no sense. Provide the average mass of the mice and the appropriate standard deviation.

Done. Lines 636 and 655.

4) Again, how do the atherogenicity and thrombogenicity indices compare to the other diets in the current study? Provide a discussion within the manuscript.

Dear reviewer, as a way of clarifying the atherogenicity and thrombogenicity indices we have modified the presentation of information in chapter 4. Materials and Methods of the article (pages 609 and 614).

We would like to clarify that our research group has been dedicated to the study of different oils within the model proposed in this article. The atherogenic index (AI) and the thrombogenic index (TI), based on the composition of free FAs, assessed the nutritional quality of oils. We have some publications that bring this information. The most recent are the following publications:

1) Marcelino, G.; Hiane, P.A.; Pott, A.; de Oliveira Filiú, W.F.; Caires, A.R.L.; Michels, F.S.; Júnior, M.R.M.; Santos, N.M.S.; Nunes, Â.A.; Oliveira, L.C.S.; Cortes, M.R.; Maldonade, I.R.; Cavalheiro, L.F.; Nazário, C.E.D.; Santana, L.F.; Di Pietro Fernandes, C.; Negrão, F.J.; Tatara, M.B.; de Faria, B.B.; Asato, M.A.; de Cássia Freitas, K.; Bogo, D.; do Nascimento, V.A.; de Cássia Avellaneda Guimarães, R. Characterization of Buriti (Mauritia flexuosa) Pulp Oil and the Effect of Its Supplementation in an In Vivo Experimental Model. Nutrients 202214, 2547. https://doi.org/10.3390/nu14122547.

2) Figueiredo, P.S.; Martins, T.N.; Ravaglia, L.M.; Alcantara, G.B.; Guimarães, R.d.C.A.; Freitas, K.d.C.; Nunes, Â.A.; de Oliveira, L.C.S.; Cortês, M.R.; Michels, F.S.; Kadri, M.C.T.; Bonfá, I.S.; Filiú, W.F.d.O.; Asato, M.A.; de Faria, B.B.; Nascimento, V.A.d.; Hiane, P.A. Linseed, Baru, and Coconut Oils: NMR-Based Metabolomics, Leukocyte Infiltration Potential In Vivo, and Their Oil Characterization. Are There Still Controversies? Nutrients 202214, 1161. https://doi.org/10.3390/nu14061161.

Thus, obtaining the fatty acid profiles through chromatography allowed us to evaluate the nutritional quality of oils according to equations of nutritional indexes. Here, AI and TI demonstrated the ability of FAs to favor or prevent atherosclerosis and coronary thrombosis, considering their effects on serum cholesterol and low-density lipoprotein (LDL) concentrations.

We discuss the results found in this study between the 370-375 lines and hope that the information is clarified. Please let us know if you still have questions regarding these results. It is possible that we can clarify.

5) This reviewer may have missed this in the first submission, but there are no units provided for Figure 5B and 5C.

We appreciate the observation. Now we have included the unit (%) in the line 243.

6) Some comments by the other reviewer could benefit with discussion points in the manuscript.

We appreciate your observation. We would like to say that the reviewer's comments have been carefully evaluated and we have made the requested adjustments, as we understand that it could enrich our manuscript. We hope to have achieved a higher quality for the text.

Finally, we really appreciate your considerations that were very helpful for the quality of our manuscript. Please let us know about any question.

Kind regards,

Karine de Cássia Freitas.

Reviewer 2 Report (New Reviewer)

The paper entitled: “Effects of the Seed Oil of Carica papaya Linn on Food Consumption, Adiposity, Metabolic and Inflammatory Profile of 2 Mice using Hyperlipidic Diet”- is in my opinion important and should be published, considering modern human problems of obesity. However, before publication I suggest some necessary corrections:

Abstract

The fatty acid composition of the oil extracted from the seeds of Carica papaya Linn was evaluated.-Please added the method of evaluation.

Under these experimental conditions, papaya seed oil led to higher amounts of monounsaturated fatty acids-It is not clear – This statement concerns the serum or the oil samples-please clarify.

Introduction

l.47-58 – I agree with these statements, but they are too obvious. I suggest removing them.

l.58 Please remove-Among the fruits.

l.58. Please rethink the following sentences:” The papaya (Carica papaya Linn) was discovered in southern Mexico and northern Nicaragua, and because it is a common plant in tropical and subtropical climates, it was brought to Brazil in 1587”- It looks like the plant was brought to Brasil because is a common plant in tropical and subtropical climates. Please rephrase this sentence.

l.67-69: “The larger the fruit, the greater the quantities of seeds it contains, and, according to Allan (1969) [7], a single papaya can produce around 1000 seeds or more, which can cause environmental problems due to the residue produced.”- Please explain what kind of problems for the environment do you mean?

l.79-84- References are needed.

In the introduction part the following ref. should be added.

Singh SP, Kumar S, Mathan SV, Tomar MS, Singh RK, Verma PK, Kumar A, Kumar S, Singh RP, Acharya A. Therapeutic application of Carica papaya leaf extract in the management of human diseases. Daru. 2020 Dec;28(2):735-744. doi: 10.1007/s40199-020-00348-7. 

Mariano LNB, Boeing T, da Silva RCV, da Silva LM, Gasparotto-Júnior A, Cechinel-Filho V, de Souza P. Exotic Medicinal Plants Used in Brazil with Diuretic Properties: A Review. Chem Biodivers. 2022 Jun;19(6):e202200258. doi: 10.1002/cbdv.202200258. 

Noor Khalidah Abdul Hamid, Peace Onas Somdare, Khadijah Abdullah Md Harashid, Nurul Ain Othman, Zulhisyam Abdul Kari, Lee Seong Wei, Mahmoud A.O. Dawood, Effect of papaya (Carica papaya) leaf extract as dietary growth promoter supplement in red hybrid tilapia (Oreochromis mossambicus × Oreochromis niloticus) diet, Saudi Journal of Biological Sciences,Volume 29, Issue 5,2022,Pages 3911-3917, https://doi.org/10.1016/j.sjbs.2022.03.004.

Patra, A., Abdullah, S. & Pradhan, R.C. Review on the extraction of bioactive compounds and characterization of fruit industry by-products. Bioresour. Bioprocess. 9, 14 (2022). https://doi.org/10.1186/s40643-022-00498-3.

Discusion

l.280-296- It should be deleted.

Table 4 should be transferred to Results.

l.317-329- This looks like a repetition of the Introduction. Please remove these sentences.

The discussion part should be definitely shortened. Please focus on your own results.

Methods

-There is a lack of details in gas chromatography (GC-MS) analysis.

-During Acute toxicity test, Authors observed among others also physical changes for 14 days. A group was supplemented with  papaya seed oil (5000 mg/kg) –this is the day or week dose?

-l.582- “After the toxicity test, the experiment was begun [74].”- The experiment is important. Please give some deteils because cited literature (Brito, A.R.S. Manual de ensaios toxicológicos “in vivo”. Ciências Médicas – Unicamp, Campinas, 1994, 15-22) isnot  in English language.

l.597- “ 1 mL/kg of animal weight”- this dose is per week or day it is not clear-please explain.

Conclusion

l.669-Remove- Based on the results found

l.672- in conclusion do not use the abbreviation

l.679_What kind of the disease do you mean? But frankly- I suggest removing completely lines:677-679. 

Author Response

Dear Reviewer 2,

We thank you for spending your time working in our manuscript. We know that these reviews are time demanding and your suggestions improved our article review.

We have modified our article according to your considerations.

The changes are listed below and marked using the “Track Changes” highlighted in yellow in the file.

Abstract

The fatty acid composition of the oil extracted from the seeds of Carica papaya Linn was evaluated. Please added the method of evaluation.

Done. Line 34.

Under these experimental conditions, papaya seed oil led to higher amounts of monounsaturated fatty acids-It is not clear – This statement concerns the serum or the oil samples-please clarify.

Done. Line 39.

Introduction

l.47-58 – I agree with these statements, but they are too obvious. I suggest removing them.

Removed lines 51 and 62.

l.58 Please remove-Among the fruits.

Removed line 64.

l.58. Please rethink the following sentences:” The papaya (Carica papaya Linn) was discovered in southern Mexico and northern Nicaragua, and because it is a common plant in tropical and subtropical climates, it was brought to Brazil in 1587”- It looks like the plant was brought to Brasil because is a common plant in tropical and subtropical climates. Please rephrase this sentence.

Done. Lines 64-68.

l.67-69: “The larger the fruit, the greater the quantities of seeds it contains, and, according to Allan (1969) [7], a single papaya can produce around 1000 seeds or more, which can cause environmental problems due to the residue produced.”- Please explain what kind of problems for the environment do you mean?

Done. Line 77-81.

l.79-84- References are needed.

Was added. Lines 100 and 803.

In the introduction part the following ref. should be added.

Singh SP, Kumar S, Mathan SV, Tomar MS, Singh RK, Verma PK, Kumar A, Kumar S, Singh RP, Acharya A. Therapeutic application of Carica papaya leaf extract in the management of human diseases. Daru. 2020 Dec;28(2):735-744. doi: 10.1007/s40199-020-00348-7. 

Was added. Lines 85 and 790.

Mariano LNB, Boeing T, da Silva RCV, da Silva LM, Gasparotto-Júnior A, Cechinel-Filho V, de Souza P. Exotic Medicinal Plants Used in Brazil with Diuretic Properties: A Review. Chem Biodivers. 2022 Jun;19(6):e202200258. doi: 10.1002/cbdv.202200258.

Was added. Lines 85 and 787.

Noor Khalidah Abdul Hamid, Peace Onas Somdare, Khadijah Abdullah Md Harashid, Nurul Ain Othman, Zulhisyam Abdul Kari, Lee Seong Wei, Mahmoud A.O. Dawood, Effect of papaya (Carica papaya) leaf extract as dietary growth promoter supplement in red hybrid tilapia (Oreochromis mossambicus × Oreochromis niloticus) diet, Saudi Journal of Biological Sciences,Volume 29, Issue 5,2022,Pages 3911-3917, https://doi.org/10.1016/j.sjbs.2022.03.004.

Was added. Lines 90 and 800.

Patra, A., Abdullah, S. & Pradhan, R.C. Review on the extraction of bioactive compounds and characterization of fruit industry by-products. Bioresour. Bioprocess. 9, 14 (2022). https://doi.org/10.1186/s40643-022-00498-3.

Was added. Lines 100 and 806.

Discusion

l.280-296- It should be deleted.

Was removed.

Table 4 should be transferred to Results.

Done. Line 299.

l.317-329- This looks like a repetition of the Introduction. Please remove these sentences.

Done.

The discussion part should be definitely shortened. Please focus on your own results.

Dear reviewer, we appreciate your suggestion as it can really make our article more interesting to the reader. Based on that, we excluded some more generic paragraphs, however, as the article has several analyzes, which brought many results, we could not reduce the discussion.

Methods

-There is a lack of details in gas chromatography (GC-MS) analysis.

Done. Lines 595-603.

-During Acute toxicity test, Authors observed among others also physical changes for 14 days. A group was supplemented with  papaya seed oil (5000 mg/kg) –this is the day or week dose?

Done. Line 639. We rewrote the paragraph for clarity.

Both the control group that was supplemented with saline solution (1 mL/kg of body weight) and the group supplemented with papaya seed oil (5000 mg/kg of body weight) received a single dose, and at times 30, 60, 120 , 240, 360 minutes and once daily for 14 days after administration, cognitive, neuromuscular and physical changes were evaluated.

-l.582- “After the toxicity test, the experiment was begun [74].”- The experiment is important. Please give some deteils because cited literature (Brito, A.R.S. Manual de ensaios toxicológicos “in vivo”. Ciências Médicas – Unicamp, Campinas, 1994, 15-22) is not  in English language.

Done. Line 640. Added test information in the text.

l.597- “ 1 mL/kg of animal weight”- this dose is per week or day it is not clear-please explain.

Done. Line 659. “1 mL/kg of animal weight per day”.

Conclusion

l.669-Remove- Based on the results found

Done. Line 736.

l.672- in conclusion do not use the abbreviation

Done. Lines 739 and 744.

l.679_What kind of the disease do you mean? But frankly- I suggest removing completely lines:677-679. 

Done. Was removed.

Finally, we really appreciate your considerations that were very helpful for the quality of our manuscript. Please let us know about any question.

Kind regards,

Karine de Cássia Freitas.

Round 2

Reviewer 1 Report (Previous Reviewer 1)

Comments have been addressed.

This manuscript is a resubmission of an earlier submission. The following is a list of the peer review reports and author responses from that submission.

Round 1

Reviewer 1 Report

In the manuscript titled “Effects of the Seed Oil of Carica papaya Linn on Food Consumption, Adiposity, Metabolic and Inflammatory Profile of Mice using Hyperlipidic Diet”, Santana et al. present information on the effect of the papaya seed oil on various outcomes. Specific comments are as follows:

1) The most important and missing item of information is the caloric content of each diet that was provided to the animals. It is very possible that a lower caloric content is present in the papaya seed oil diet, which could then translate into lower caloric intake. Are the oil-supplemented high-fat diets isocaloric? If not, the study is questionable on the effects the oil exhibits when compared with other oils.

2) The data within Table 2 make no sense. First, there are no units provided, but second the data are duplicated between the ‘Food Intake’ and ‘Caloric Consumption’ headings.

3) There are significant issues with formatting of the document. For Table 3, the decimal point error data are found in many instances on a second line. Also, parts of Figures 2 and 3, and Table 4 are off the page. These issues may be from the formatting by the journal, but this should have been caught by the authors before approval.

4) In what little this reviewer can discern from Figure 2, it seems that the cells’ area in panel F are comparable to that with panel D; however, while axes are completely missing in Figure 3, if the order of the bars are the same as that for the images, the OS and OM (papaya) diets show a sharp difference. This is confusing, as these data should be statistically significant given the standard deviations and n values.

5) The abstract differs compared to the main text, such that the abstract indicates no difference in food intake with the papaya oil diet, whereas the main text states there is a difference (but again, Table 2 makes no sense).

6) For the animals, in section 4.5 what is meant by “adults (+/- 16g)”? Were animals kept in metabolic cages? Were animals in groups or in individual cages? How animals were housed can influence the outcomes, and especially if animals were (or were not) in metabolic cages.

7) How do the atherogenicicy and thrombogenicity indices compare versus the other diets in the study?

Reviewer 2 Report

General comment:

Because of too many problems, it is difficult to evaluate the significance, novelty and impact of this study.

Specific comments

1) Detailed information of dietary composition such as CT, AIN-93, HPL, etc. must be clearly indicated in Table.

2) As comparison, the diet groups of supplementation of soybean oil and olive oil were conducted. However, the diet group of saturated fatty acid-rich fat such as beef tallow or lard is also essential to evaluate the functional benefits of Carica pappaya Linn oil as comparison.   

3) Already, the composition of fatty acids (including oleic acid) in Carica pappaya Linn has been reported by other groups. The authors must describe what is the originality or novelty of the data of fatty acids indicated.

4) There are too many type problems.  

5) Line 660, 672: What is "± 16g"?

6) Line 661: "I mL/kg"  Is this "1 mL/kg of body wt per a day" or 1 mL/kg diet ?